# Coherence-based Query Performance Measures for Dense Retrieval

## ABSTRACT

Query Performance Prediction (QPP) estimates the effectiveness of a search engine's results in response to a query without relevance judgments. Traditionally, *post-retrieval* predictors have focused upon either the distribution of the retrieval scores, or the coherence of the top-ranked documents using traditional bag-of-words index representations. More recently, BERT-based models using dense embedded document representations have been used to create new predictors, but mostly applied to predict the performance of rankings created by BM25. Instead, we aim to predict the effectiveness of rankings created by single-representation dense retrieval models (ANCE & TCT-ColBERT). Therefore, we propose a number of variants of existing unsupervised coherence-based predictors that employ neural embedding representations. In our experiments on the TREC Deep Learning Track datasets, we demonstrate improved accuracy upon dense retrieval (up to 92% compared to sparse variants for TCT-ColBERT and 188% for ANCE). Going deeper, we select the most representative and best performing predictors to study the importance of differences among predictors and query types on query performance. Using existing distribution-based evaluation QPP measures and a particular type of linear mixed model, we find that query types further significantly influence query performance (and are up to 35% responsible for the unstable performance of QPP predictors), and that this sensitivity is unique to dense retrieval models. In particular, we find that in the cases where our predictors perform lower than score-based predictors, this is partially due to the sensitivity of MAP@100 to query types. Our novel analysis provides new insights into dense QPP that can explain potential unstable performance of existing predictors and outlines the unique characteristics of different query types on dense retrieval models.

**ACM Reference Format:**

Anonymous Author(s). 2024. Coherence-based Query Performance Measures for Dense Retrieval. In *2024 ACM SIGIR International Conference on the Theory of Information Retrieval (ICTIR '24), July 13, 2024, Washington D.C., USA.* ACM, New York, NY, USA, 11 pages. https://doi.org/10.1145/nnnnnnn.nnnnnnn

## 1 INTRODUCTION

Retrieval effectiveness in search engines can vary across different queries [22, 53]. Being able to accurately predict the likely effectiveness of a search engine for a given query may facilitate interventions, such as asking the user to reformulate the query [5, 29, 40, 54].

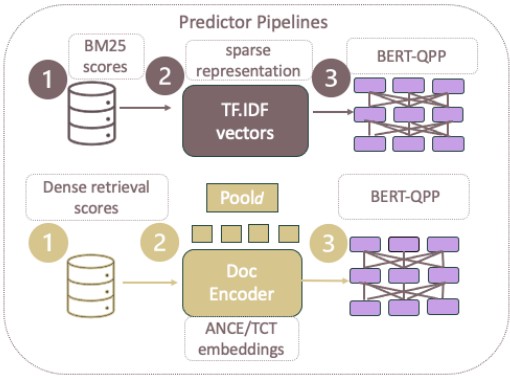

**Figure 1: Schematic representation of recent QPP pipelines, together with our proposed approach (Step 2, bottom). Top: A BM25 ranking consisting of TF.IDF vector representations (Step 2) [1, 16], and fine-tuning BERT-based models on top of existing rankings (Step 3) [2, 14, 23, 58]. Bottom: Dense retrieval ranking with dense embedded representations. Numbers denote each step in the pipeline.**

To this end, the task of *Query Performance Prediction (QPP)* aims to predict the effectiveness of a search result in response to a query without having access to relevance judgments [7]. In the last two decades, a number of *query performance predictors* have been proposed, which can be grouped in two main categories: *Pre-retrieval* predictors estimate query performance using only linguistic or statistical information contained in the queries or the corpus [24, 25, 34, 46, 59]. On the other hand, *post-retrieval* predictors use the relevance scores or contents of the top returned documents, by measuring, for example, the focus of the result list compared to the corpus [11, 60], or the distribution of the scores of the top-ranked documents [12, 39, 43, 47, 51]. Predictors based on NQC [49] (the standard deviation of relevance scores) have been found to be surprisingly accurate. A further group of predictors examine the pairwise similarities among the retrieved documents [1, 16]. Thus far, these predictors have been applied using traditional bag-of-words representations. While examining the coherence between returned documents is useful, as we show, these representations are not suitable for predicting the query performance of more advanced retrieval methods.

More recently, pre-trained language models (PLMs) have introduced neural network architectures that encode the embeddings of queries and documents [15, 27, 28, 56], and have led to increased retrieval effectiveness. Often, a BERT-based model is trained for use as a reranker of the result retrieved by (e.g.) BM25 [41] - such *cross-encoders* include BERT_CLS [36] and monoT5 [37]. On the other hand, *dense retrieval approaches* [26, 56] are increasingly popular, whereby embedding-based representations of documents are indexed, and those with the similar embeddings to the query are

identified through nearest-neighbour search (e.g. ANCE [56], TCT-ColBERT [28]). Compared to reranking setups, dense retrieval is attractive as recall is not limited by the initial BM25 retrieval approach, and improvements in the PLM can improve all aspects of the retrieval effectiveness. Therefore, dense retrieval models inspire us to develop predictors that are effective for predicting their rankings.

In parallel, neural architectures have also been adopted as methods for predicting query difficulty. These post-retrieval methods are *supervised*, and use refined neural architectures in order to produce a final performance estimate [2, 14, 23, 58]. For instance, BERT-QPP [2] fine-tunes BERT [15] embeddings for QPP by estimating the relevance of the top-ranked document retrieved for each query. However, its performance is lower or outperformed by unsupervised predictors when using advanced retrieval methods and the TREC Deep Learning datasets [18]. In our view, the problem lies in the mismatch of representations between predictor and ranking, which is best described in Figure 1. On top, we see the pipeline resulting from a BM25 ranking, and, at the bottom, a ranking from a dense retrieval system [26, 56]. While BERT-based QPP techniques can be used to predict the effectiveness of BM25 [2, 14, 23, 58], single-representation dense retrieval models already contain representations that can accurately predict their corresponding ranking, thus eliminating the need to apply step 3 (BERT-QPP). Instead, to create predictors applicable for dense retrieval, we could use the existing embedded representations (step 2). Indeed, by considering patterns among the embeddings of the retrieved documents, we can update existing unsupervised predictors from traditional sparse [1, 16] to dense representation-based.

At the same time, the selection of evaluation measure can have an impact on the conclusions of QPP experimental results. This observation is more prominent if we consider, for example, that unsupervised QPP predictors such as NQC [47] were primarily optimised for MAP at deeper cutoffs (100 or 1000); on the other hand, more recent supervised predictors were either optimised for RR@10 [2, 23] or used both NDCG@10 and RR@10 [14] providing comparable results between the two measures, but in both cases, results for MAP were missing. As a result, it is impossible to provide insights that are fully generalisable, as missing to report either of them can lead to biased results and incomplete conclusions. We believe that designing experimental studies should be aligned with the idea that the different measures are not interchangeable, and that proposed predictors could be complemented with the case where the predictor fails, together with the explanation of the reasons why this happens.

One explanation could be that query performance is further mediated by query categorisation. Few works have examined how QPP varies with query categories [8, 19]. Indeed, knowing which queries are more difficult to answer may inform us about how to develop more refined predictors. In a recent query taxonomy [6], certain categories were found to be more difficult to answer compared to others. Therefore, we also quantify the extent to which query categories are responsible for the unstable performance of QPPs across different evaluation measures.

In short, our contributions are the following: (i) We propose a number of embedding variants of existing coherence predictors and our own extension *pairRatio*, an unsupervised predictor which uses pairwise relations of embedding vectors. In this way, we create predictors designed for dense retrieval; (ii) We study existing predictors to two state-of-the-art single-representation dense retrieval models,

namely ANCE [56] and TCT-ColBERT [28], as well as BM25 using all three evaluation metrics currently used for QPP, and show that changing the representations increases performance significantly not just for dense but also sparse retrieval; (iii) By also comparing with supervised predictors, we show that applying a BERT-based model for dense QPP is an unnecessary step in the pipeline that decreases QPP performance; (iv) We apply multilevel statistical models [13, 21, 32, 50] in QPP to quantify the relationship between query categorisation and the unstable QPPs. In our analyses, we measure the performance of different QPPs in relation to the total QPP variation that can be attributed to the categorisation or as we term *query types*. At the same time, we detect a unique sensitivity of dense retrieval methods, which are affected by query type (up to 35% increase in query performance variations due to query categorisation) and exhibit larger differences between predictors, a pattern which is not apparent in sparse retrieval.

In addition, we observe: (a) Our proposed predictors provide the highest correlations for the more precision-oriented NDCG@10 for all retrieval models, while NDCG@10 and MRR@10 provides similar results. (b) Our multilevel perspective proposes a solution to correlation instabilities between measures, by showing how the interplay with query types differently influences each of the measures. In other words, we provide an analytical point that can explain any predictor, and show how our proposed predictors mainly optimise the measure that is less influenced by query variations. The structure of the rest of this paper is as follows: We present related work in Section 2, and present our new extended predictors in Section 3. Then, we follow with traditional correlation analysis of QPP predictors in Sections 4 and 5, continue with an extended linear mixed model analysis to test for query type in Section 6, and conclude with some final remarks in Section 7.

## 2 RELATED WORK

The focus of this paper is on post-retrieval QPPs, as they are in general more accurate than pre-retrieval QPPs [24]. Indeed, there are two main reason why we eliminate pre-retrieval predictors from our focus. First, existing unsupervised neural pre-retrieval predictors [3, 44, 45] propose, for example, geometric semantic similarities of query terms, which indicate query specificity or contextual similarity and are based on pre-trained neural embeddings. Since these predictors examine queries at the token-level, they are not applicable to single-representation dense retrieval. Second, information based on queries can, in general, provide quite limited information with respect to the effectiveness of the ranking.

In terms of post-retrieval QPP, earlier post-retrieval predictors examined the focus on the result list induced by language models (probability distributions of all single terms) [11]. For example, *Clarity* [11] measures the divergence of the language model of top-ranked documents from the one of the corpus(irrelevant list) - the higher the divergence, the better the performance. *Utility Estimation Framework (UEF)* [48] uses pseudo-effective reference lists induced by term probability-based language models and estimates their relevance using predictors such as NQC (see below in Section 2.1). Both of these rely upon term probabilities, and are, therefore, not feasible for extending our predictions to dense retrieval. *Query Feedback (QF)* [60] refers to the overlap of the returned documents with those obtained after applying pseudo-relevance feedback - yet,

pseudo relevance feedback approaches for dense retrieval are still in their infancy [55, 57], so we do not consider QF further.

In the remainder of this section, we discuss the main types of query performance predictors that could be applied to dense retrieval, specifically score-based unsupervised predictors (Section 2.1) and document representation-based predictors (Section 2.2).

## 2.1 Score-based QPP

Score-based predictors encode certain assumptions about how the scores should be distributed for high or low-performing queries. For instance, a simple predictor might be the *Maximum Score* among the retrieved documents [42] - the higher the maximum score, the more confident the retrieval system is that it has found a document that matches well the query. The most commonly applied score-based predictor is *Normalised Query Commitment (NQC)* [47], based on the standard deviation of the retrieval scores, which is negatively correlated with the amount of query drift (the non-related information in the result list) [33]. Several variations of NQC have been proposed that further enhance its accuracy [12, 39], incorporate the scores magnitude [51], or estimate a more robust version of variance with bootstrapping. Indeed, *Robust Standard Deviation estimator (RSD)* [43] extends NQC results to multiple contexts (each with a bootstrap sample) representing a population of scores [43]. Score-based predictors (step 1 in Figure 1) are easily applicable to dense retrieval, since scores are computed by each retrieval method.

## 2.2 Document Representation-based QPP

Predictors based on document representations [1–3, 14, 16, 17, 23, 44, 45] capture semantic relations either between queries, documents, or their interaction [15, 28] - we discuss unsupervised and supervised predictors below.

*2.2.1 Unsupervised Coherence Predictors.* In general, effective unsupervised predictors that consider document representations are preferable, since they require less computation than supervised predictors. One example of an unsupervised predictor that examines the lexical representations of documents is *spatial autocorrelation* [16], which considers the spatial proximity of lexical document representations, by using their pairwise TF.IDF-based similarities to produce a new set of scores "diffused in space". The final predictor is obtained by correlating the original scores with the diffused scores. Indeed, a low correlation between scores of topically-close documents is assumed to imply a poor retrieval performance.

Another family of recent coherence-based predictors creates a graph of the most similar documents among the top-ranked documents [1], based on their TF-IDF representations. Specifically, metrics such as Weighted Average Neighbour Degree (WAND) and Weighted Density (WD) were found to enhance the performance of score-based predictors after linear interpolation. These predictors (applied step 2 in Figure 1, top) were proposed for sparse document representations and have not previously been applied to dense embedded representations.

*2.2.2 Supervised & Neural Predictors.* In general, supervised models for QPP can be attractive due to the varying sources of indicators for inferring query performance [42]. At the same time, they are computationally complex compared to unsupervised predictors. For example, Neural-QPP [58] is a multi-component supervised predictor as the output of existing unsupervised QPP predictors with weak supervision - we can think of this as a neural supervised

aggregation predictor. More recently, BERT-QPP [2] fine-tunes a BERT model for the QPP task by adding cross-encoder or bi-encoder layers that estimate an effectiveness measure (e.g. NDCG) based on the contents of the top returned document in response to the query. While BERT-QPP can also be applied to the dense retrieval rankings, it uses a different model to that used by the dense retrieval approach itself. Out of the two BERT-QPP variants, the bi-encoder version is closer to the intuition of single-representation dense retrieval. Finally, qppBERT-PL [14] adds an LSTM network on top of the BERT representation to model both document contents and the progression of estimated relevance in the ranking. Compared to BERT-QPP, this approach has promise as it considers more information than just the top-ranked document.

To summarise, existing predictors have either focused on sparse document representations or retrieval scores on the unsupervised side, or have introduced neural pre-trained architectures to create more complex supervised predictors. However, no work has addressed unsupervised predictors using dense embedded representations, as are readily available in dense retrieval configuration. Instead, we argue that by using simple predictors that consider document representation resulting from dense models (step 2 of Figure 1, bottom), we can accurately predict effectiveness without the need for supervised cross-encoder-based methods (step 3). In the next section, we detail existing predictors that can be applied to dense retrieval.

## 3 COHERENCE PREDICTORS FOR DENSE RETRIEVAL

In this section, we first describe some existing sparse coherence-based predictors in Section 3.1, and then show how these can be adapted to be better suited for dense retrieval settings in Section 3.2.

## 3.1 Sparse Coherence-based Methods

*3.1.1 Spatial Autocorrelation (AC).* First, consider $d$ to be a document's TF.IDF vector. Then, the inner product of two documents at ranks $i$ and $j$ is given by $sim(d_i, d_j)$. We can obtain a pairwise similarity matrix among $k$ top-ranked documents as follows:

$$W = \begin{bmatrix} sim(d_{11}) & sim(d_{12}) & ... & sim(d_{1k}) \\ ... & ... & ... & ... \\ sim(d_{k1}) & sim(d_{k2}) & ... & sim(d_{kk}) \end{bmatrix} \quad (1)$$

where $k$ is the cutoff number of the top-k documents. For brevity of notation, let $sim(d_{ij}) = sim(d_i, d_j)$. Projecting (multiplying) each element of the matrix $W_{ij}$ on the vector of the original retrieved scores, $Score(\vec{d})$, we can obtain a new list of *diffused* scores as:

$$Score(\tilde{d}) = W * Score(d) \quad (2)$$

Thereafter, an estimate of the spatial autocorrelation (AC) [16] is obtained by using the Pearson correlation between the two vectors:

$$AC = corr(Score(\tilde{d}), Score(d)) \quad (3)$$

which quantifies the relation between the initial and diffused scores. Indeed, as mentioned above, a low correlation between the original retrieval scores (i.e. $Score(d)$) and those weighted by their topical similarity (the diffused scores, $Score(\tilde{d})$) was found to imply poor retrieval performance [16].

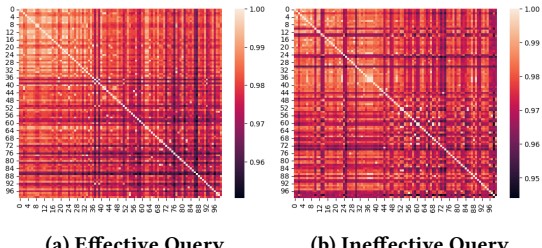

**(a) Effective Query**    **(b) Ineffective Query**

**Figure 2: Heatmap of pairwise similarity matrix of the top-100 TCT-ColBERT document embeddings for returned for the best (query id 104861 with NDCG@10=1) and worst performing queries (query id 489204 with NDCG@10=0.189) from the TREC DL 19 queryset.**

*3.1.2 Network Metrics.* As mentioned above, the matrix $W$ represents all pairwise similarities between the top-retrieved documents. This matrix is equivalent to a fully connected network, where each node $V_G$ corresponds to the $d$ TF.IDF vector, and each edge $E_G$ corresponds to each entry $sim(d_{ij})$ [1], or more formally $G(q, D_q^{(k)}) = \{V_G, E_G, W\}$. In this regard, to avoid all edges being considered equal without attention to the edge weight, the network is further pruned via thresholding [9], where the similarities higher than the mean similarity value are selected as neighbours.

Consequently, we have the following definitions, which correspond to some recently proposed network metrics [1] for QPP:

$$AverageNeighbourDegree(AND) = \frac{1}{k}\sum_{i=1}^{k}(\frac{1}{|N_{d_i}|}\sum_{j \in N_{d_i}}|N_{d_j}|) \quad (4)$$

where $N_{d_i}$ is the neighbourhood of document $d_i$. Typically, Equation (4) is applied on the pruned graph that only contains edges between the most similar documents, and hence corresponds to the more accurate *Weighted AND* (WAND) measure [1].

Another way to think about coherence is to count the observed edges or similarities over the set of all possible edges. This results in the Density measure, as follows:

$$Density(D) = \frac{2|E_G|}{|V_G|(|V_G| - 1)} \quad (5)$$

In short, a higher neighbourhood degree and a higher density of a graph network indicates a more coherent set of top-retrieved results. The general intuition behind these measures is that the presence of coherence, as reflected by highly similar documents in a top-retrieved set indicates the ability of the retrieval method to distinguish relevant from non-relevant documents, and therefore, return the relevant ones at the top of the list.

## 3.2 Dense Coherence-based Methods

We now derive the embedding representation variants of the above predictors in order to make them suitable for the prediction of neural dense retrievers. We first create the variants for embedding-based AC and network metrics, and then introduce a new variant that extends AC by considering rank groupings.

*3.2.1 AC-embs.* Let $\phi_d$ and $\theta_q$ respectively represent the dense embedded representation of a document and a query. Firstly, we adapt autocorrelation, such that instead of TF.IDF vectors we consider the embedded document representations. Let the inner product of

two documents at ranks $i$ and $j$ (with embeddings $\phi_i$ and $\phi_j$) be written $sim(\phi_{dij})$, then we can define the pairwise similarities of the top $k$ ranked documents as:

$$W^\phi = \begin{bmatrix} sim(\phi_{d11}) & sim(\phi_{d12}) & ... & sim(\phi_{d1k}) \\ ... & ... & ... & ... \\ sim(\phi_{dk1}) & sim(\phi_{dk2}) & ... & sim(\phi_{dkk}) \end{bmatrix} \quad (6)$$

We can then apply autocorrelation (denoted as AC above) as per Equations (2) & (3). We denote this as *AC-embs*.

*3.2.2 Network-embs.* Similarly, and as we showed that the similarity matrix is equivalent to a fully connected network set of edges, we can apply WAND and WD as per Equations (4) & (5), denoted as *WAND-embs* and *WD-embs*, respectively.

*3.2.3 pairRatio.* We now introduce an extension of AC-embs inspired by visually exploring embedding relations. Specifically, in Figure 2, we visualise the pairwise similarity matrix ($W_\phi$) obtained using TCT-ColBERT [28] embeddings for the top-100 passages for the one high and one low performing query in the TREC Deep Learning Track 2019 queryset. For the best performing query, there is higher pairwise similarity among documents of top ranks (top left corner, indicated by a group of lighter shading), and lower correlation for lower ranks (darker shading). On the other hand, for the worst query, elements of darker shading appear at high ranks, indicating that the top-ranked documents may not be as coherent). In addition, there is less dark shading in low ranks compared to the best query. These observations inspire us to explore the trend of average top vs. bottom rank pairwise similarities of top-ranked embeddings.

Specifically, let $W^\phi_{\tau_1..\tau_2}$ denote the (diagonal) subset of $W^\phi$ between ranks $\tau_1$ and $\tau_2$. Then, for a given rank threshold $\tau$, we can measure the ratio between the mean pairwise similarity above and below rank $\tau$, i.e. $W^\phi_{0..\tau}$ and $W^\phi_{\tau..k}$ as follows:

$$pairRatio(W^\phi) = (\overline{W^\phi_{1..\tau_i}}) \cdot (\overline{W^\phi_{\tau_j..k}})^{-1} \quad (7)$$

where $\overline{W^\phi}$ denotes the mean of the given matrix, $\tau_i$ corresponds to the end of the upper matrix, and $\tau_j$ symbolises the start of the lower matrix (we use the two cutoff points as separate hyperparameters). We called this predictor *pairRatio*. Unlike WAND and WD, we consider the magnitude of this contrast as indicative of query performance. We believe that, since this relates to the retrieval method itself, it should be indicative of query performance especially for advanced retrieval methods.

Still, the similarity matrix $W^\phi$ can only provide information about the relative similarity of documents. Introducing some information about the document scores would increase performance prediction accuracy, since it relates to the absolute ranking of each document. Let $A$ be an adjusted matrix, where each entry, for a document pair $i$ and $j$ is multiplied by the final similarity of the query to each of the documents:

$$A_{ij} = W_{ij} \cdot (\phi_i \cdot \theta_q) \cdot (\phi_j \cdot \theta_q)$$

$A$ better encodes similarity of the query among the pairwise document similarities. pairRatio (Equation (7)) can then be applied upon $A$, which we denote as adjusted pairRatio, or *A-pairRatio*.

In short, we are interested in the effectiveness of these predictors based on dense document representations and how they perform

in relation to their sparse versions. We test their performance compared to score-based and supervised predictors in Section 5.

## 4 EXPERIMENTAL SETUP

Our experiments address the following research questions:

**RQ1** How do unsupervised coherence-based predictors compare to unsupervised score-based predictors in dense and sparse retrieval?

**RQ2** How do unsupervised predictors perform compared to supervised predictors in dense and sparse retrieval?

To address these research questions, our setup is as follows:

**Datasets:** We use the MSMARCO passage ranking corpus, and apply the TREC Deep Learning track 2019 and 2020 query sets, containing respectively 43 and 54 queries with relevance judgements. In particular, each query in these querysets contains many judgements obtained by pooling various distinct retrieval systems.

**QPP Predictors**: As unsupervised score-based predictors, we apply Max score (MAX) [42], and NQC [47]. As a representative variant of NQC, we choose *RSD*. This bootstrap-based predictor is the most recent NQC variant and was shown to outperform other score-based predictors. Specifically, we use the *RSD(uni)* version which samples documents uniformly. For each cutoff, we sample from 0.60 to 0.80 of the initial result list size. We use spatial auto-correlation (AC) [16], WAND and WD [1], and the interpolation of WAND and WD with NQC (following the findings of the original paper [1], which suggest that network metrics further increase the performance of NQC). We also report our embedding variants (AC-embs, WAND-embs, WD-embs, PairRatio, A-PairRatio). For each unsupervised predictor, we tune the hyperparameters of each dataset on the other. Specifically, to tune the cutoff value for the top-$k$ documents all unsupervised predictors including ours, we use a grid of values [5,10,20,50,100,200,500,1000]. For PairRatio and A-PairRatio, we also vary the other upper and lower rank threshold hyperparameters $\tau_i$ and $\tau_j$.

For supervised predictors, we report the bi-encoder and cross-encoder variants of BERT-QPP [2]. To achieve this, we retrained the BERT-QPP cross-encoder and bi-encoder models specifically for each of the dense retrieval models. These supervised predictors exhibit their highest correlations mainly for MRR, which means that they train models that estimate the relevance of the top document of a ranking. In this regard, we check whether an alternative supervised predictor (which we call *top-1(monoT5)*) that uses only the top-retrieved document to a monoT5 model [37] – i.e. trained for relevance estimation and ranking rather than performance prediction – can perform well in dense retrieval. Note that we deliberately use the term *QPP Predictors* instead of *baselines*, since our purpose is not to demonstrate the superiority of a single predictor, but rather how a *group of predictors* behaves under different contexts and retrieval models.

**Retrieval Systems**: We deploy three retrieval approaches: BM25 sparse retrieval (applying Porter's English stemmer and removing standard stopwords) as implemented by Terrier [38], and two single-representation dense retrieval approaches, namely ANCE [56], and TCT-ColBERT [28] with PyTerrier [30] integrations.[1]

---

[1]https://github.com/terrierteam/pyterrier_dr

**Measures**: Following the TREC 2019 Deep Learning Track Overview [10], we measure system effectiveness in terms of NDCG@10 and MAP@100. We further add MRR@10, following some recent work [2, 23]. To quantify the accuracy of the QPP techniques, we adopt Kendall's $\tau$ correlation measure, as typically reported in QPP literature [11, 16, 24, 47–49, 58].[2]

## 5 CORRELATION RESULTS

Tables 1 and 2 show the accuracy of all our examined predictors on the TREC DL 2019 and 2020 query sets, respectively. Within each table: groups of columns denote the various retrieval approaches; the uppermost row reports the mean effectiveness of each ranking approach for each evaluation measure; the next group of rows contains the Kendall's $\tau$ correlation of the score-based predictors, the next one the unsupervised lexical coherence-based predictors; then we report the results for the embedding-based predictors; and finally for the supervised predictors [2].

### 5.1 RQ1: Score-based vs Coherence-based Predictors

As expected, for BM25, distribution-based score predictors (NQC and RSD(uni) show high accuracy for MAP@100 and NDCG@10, while their accuracy is lower for MRR@10, especially for DL 19. However, unlike older datasets, sparse coherence predictors are very low for TREC DL datasets. As for dense coherence predictors, surprisingly, AC-embs variant is the best performing predictor for AP@100, and for NDCG@10 on 2020. As for our pairRatio variants, they are less effective than other unsupervised predictors, such as NQC and AC-embs (except for MRR@10), as well as supervised predictors on MRRR@10.

Next we consider the two dense retrieval settings, i.e. ANCE & TCT-ColBERT. For TCT-ColBERT, we observe that our pairRatio predictors outperform not only supervised predictors, but also NQC (the best performing unsupervised predictor) for NDCG@10 and MRR@10 for both datasets, are only behind RS(uni) for MRR@10 in the DL 2019 dataset, and are competitive for AP@100. Another observation is that A-pairRatio has increased the accuracy compared to pairRatio, particularly for the TCT-ColBERT model, which indicates the need for including document-query relations. In summary, for NDCG@10 and MRR@10, for TREC DL 2020, in all four cases our dense coherence-based predictors (any of them considered) outperform score-based predictors; for TREC DL 2019, in two of the four cases ours are higher, in one case RSD is higher, and in one case they are identical. For ANCE, WAND-embs and WD-embs are better than score-based predictors for NDCG@10 and MRR@10 for the 2020 datatset, while they are only slightly behind them in the 2019 dataset. Overall, for MAP@100, NQC or RSD (uni) consistently outperform coherence-based predictors, while for NDCG@10 and MRR@10, the picture is more unstable; however, in most cases, coherence-based predictors win for dense retrieval. Further, as might be expected, changing the type of representations from sparse to dense increases the performance of coherence-based predictors across the dense retrieval settings (for ANCE,in 7 out of

---

[2]In general, Kendall's $\tau$ gives lower scores than Pearson's and Spearman's correlation, but makes the least assumptions about a linear relationship between variables. We prefer to use the space for reporting three evaluation measures.

**Table 1: Kendall's $\tau$ correlations of unsupervised and supervised predictors for TREC DL 2019. The highest correlation by an unsupervised predictor in each column is emphasised in bold and (*) indicates significance at $\alpha = 0.05$.**

| | BM25 | | | ANCE | | | TCT | | |
|---|---|---|---|---|---|---|---|---|---|
| | MAP@100 | NDCG@10 | MRR@10 | MAP@100 | NDCG@10 | MRR@10 | MAP@100 | NDCG@10 | MRR@10 |
| Effectiveness | 0.232 | 0.479 | 0.639 | 0.332 | 0.643 | 0.806 | 0.402 | 0.720 | 0.898 |
| | | | | Score-based | | | | | |
| Max | 0.171 | 0.157 | 0.087 | 0.428* | 0.316* | 0.241* | 0.297* | 0.250* | 0.015 |
| NQC | 0.322* | 0.281* | 0.075 | **0.499*** | 0.463* | 0.216 | **0.335*** | 0.243* | 0.171 |
| RSD(uni) | 0.328* | 0.288* | 0.077 | 0.495* | **0.467*** | 0.264* | **0.335*** | 0.228* | **0.227** |
| | | | | Sparse Coherence-based | | | | | |
| AC | 0.156 | 0.073 | 0.071 | 0.111 | 0.081 | 0.061 | 0.080 | -0.198 | -0.051 |
| WAND | 0.209* | 0.126 | 0.111 | 0.187 | 0.113 | 0.025 | 0.189 | 0.095 | -0.006 |
| WD | 0.158 | 0.101 | 0.087 | 0.158 | -0.004 | -0.009 | 0.184 | 0.121 | 0.015 |
| WAND(NQC) | 0.258* | 0.148 | 0.124 | 0.178 | 0.113 | 0.025 | 0.189 | 0.095 | -0.01 |
| WD(NQC) | 0.200* | 0.186 | 0.035 | 0.158 | -0.008 | -0.012 | 0.180 | 0.135 | 0.006 |
| | | | | Dense Coherence-based | | | | | |
| WAND-embs | -0.096 | -0.232 | -0.019 | 0.138 | -0.157 | -0.029 | -0.036 | 0.139 | 0.041 |
| WD-embs | 0.224* | -0.170 | 0.014 | 0.089 | -0.219 | -0.241* | -0.147 | -0.033 | 0.045 |
| AC-embs | 0.373* | 0.144 | 0.098 | 0.437* | 0.285* | 0.261* | 0.056 | 0.018 | -0.129 |
| pairRatio(ours) | 0.171 | 0.270* | 0.194 | 0.295* | 0.334* | 0.087 | 0.200 | 0.248* | -0.060 |
| A-pairRatio(ours) | **0.446*** | **0.352*** | 0.142 | 0.382* | 0.403* | 0.216 | 0.280* | **0.259*** | 0.171 |
| | | | | Supervised | | | | | |
| BERT-QPP (bi) | 0.229* | 0.305* | 0.260* | 0.162 | 0.144 | 0.067 | 0.111 | 0.048 | 0.083 |
| BERT-QPP(cross) | 0.264* | 0.254* | 0.174* | 0.198 | 0.117 | 0.038 | 0.211* | 0.088 | 0.041 |
| top-1(mono-T5) | 0.180 | 0.294* | **0.359*** | 0.224* | 0.294* | **0.470*** | 0.058 | 0.038 | 0.086 |

**Table 2: Results on TREC DL 2020. Notation as per Table 1.**

| | BM25 | | | ANCE | | | TCT | | |
|---|---|---|---|---|---|---|---|---|---|
| | MAP@100 | NDCG@10 | MRR@10 | MAP@100 | NDCG@10 | MRR@10 | MAP@100 | NDCG@10 | MRR@10 |
| Effectiveness | 0.275 | 0.493 | 0.614 | 0.363 | 0.607 | 0.803 | 0.454 | 0.686 | 0.831 |
| | | | | Score-based | | | | | |
| Max | 0.215* | 0.214* | 0.184 | 0.213* | 0.285* | 0.337* | 0.342* | 0.243* | 0.062 |
| NQC | 0.526* | 0.438* | 0.281* | **0.443*** | 0.082 | 0.172* | **0.454*** | 0.246* | 0.133 |
| RSD(uni) | 0.568* | 0.431* | 0.288* | 0.403* | 0.275* | 0.155 | 0.335* | 0.341* | 0.208* |
| | | | | Sparse Coherence-based | | | | | |
| AC | -0.199* | 0.017 | -0.097 | -0.115 | -0.022 | -0.014 | 0.018 | -0.118 | 0.030 |
| WAND | 0.189* | -0.031 | -0.026 | 0.130 | 0.009 | -0.065 | 0.208* | 0.220* | 0.023 |
| WD | 0.183* | 0.006 | -0.036 | 0.158 | 0.044 | 0.010 | 0.225* | 0.216* | 0.018 |
| WAND(NQC) | 0.220* | 0.101 | -0.024 | 0.130 | 0.005 | -0.067 | 0.202* | 0.213* | 0.188 |
| WD(NQC) | 0.253* | 0.160 | 0.036 | 0.148 | 0.023 | -0.010 | 0.223* | 0.192* | 0.004 |
| | | | | Dense Coherence-based | | | | | |
| WAND-embs | 0.038 | 0.137 | 0.042 | 0.291* | **0.300*** | 0.077 | -0.05 | 0.107 | -0.066 |
| WD-embs | 0.099 | 0.158 | 0.028 | 0.213* | 0.289* | **0.394*** | 0.127 | 0.127 | -0.161 |
| AC-embs | **0.607*** | **0.443*** | 0.339* | 0.324* | 0.219* | 0.149 | 0.121 | 0.137 | -0.002 |
| pairRatio(ours) | 0.271* | 0.203* | 0.130 | 0.178 | 0.186 | -0.132 | 0.364* | 0.318* | **-0.280*** |
| A-pairRatio(ours) | 0.482* | 0.316* | 0.189 | 0.348* | 0.270* | 0.115 | 0.429* | **0.363*** | -0.244* |
| | | | | Supervised | | | | | |
| BERT-QPP (bi) | 0.322* | 0.315* | 0.351* | 0.274* | 0.047 | 0.058 | 0.353* | 0.195* | 0.083 |
| BERT-QPP(cross) | 0.375* | 0.345* | 0.403* | 0.180 | 0.043 | 0.012 | 0.261* | 0.173 | 0.041 |
| top-1(mono-T5) | 0.371* | 0.400* | **0.534*** | 0.259* | 0.237* | 0.365* | 0.279* | 0.240* | 0.159 |

9 (QPP, Measure) cases in TREC 2019, and 9 out of 9 for TREC 2020; for TCT-ColBERT, our pairRatio variants are more effective), as the updated representations match those of the retrieval methods. To answer RQ1, for dense retrieval, score-based predictors perform

well for MAP@100, while coherence-based predictors show increased accuracy for NDCG@10 and MRR@10. For sparse retrieval, dense coherence predictors are in general better than score-based predictors.

## 5.2 RQ2: Unsupervised vs. Supervised Predictors

Next, we compare the performance of unsupervised with supervised QPP predictors for each retrieval method. For BM25, we are able to reproduce the results of the bi-encoder and cross-encoder variants of BERT-QPP, as reflected by the higher values in MRR and the competitive correlation on the other two metrics. For BM25, we used the authors' checkpoints, while we re-trained the method for ANCE & TCT-ColBERT. However, their values are still lower than NQC, (a simple score-based unsupervised predictor), and RSD(uni) (NDCG@10 on the TREC 2019 queryset), our pairRatio (MRR@10 on the 2019 queryset), AC-embs (AP@100 on 2019, AP@100 on 2020, NDCG@10 on 2020), and top-1 monoT5 (MRR@10 on both datasets). Most importantly, for the two dense retrieval methods, supervised predictors are not as effective as unsupervised predictors, such as Max and NQC. For TCT-ColBERT, supervised predictors are less effective than our pairRatio variants for NDCG@10 and MRR@10, and NQC and RSD(uni) for all metrics. The strongest observed correlations of BERT-QPP variants in dense retrieval are for AP@100. However, they have a cost to deploy (applying a BERT model on the top-ranked result). We argue that this resource would be better spent to re-rank the top results. In addition, the simpler "supervised" variant, *top-1(mono-T5),* which uses the monoT5 score of the top-ranked document is a more accurate predictor than BERT-QPP across all retrieval methods, particularly for MRR@10, which is the metric that BERT-QPP is most competitive. This surprising result shows that BERT-QPP is itself just a relevance estimator for the top-ranked document that has been trained to predict MRR@10; using any effective relevance estimator can do as good a job, if not better. To answer RQ2, we find that the existing BERT-QPP supervised predictors are less accurate than unsupervised predictors (existing and ours) for dense retrieval.

## 6 MODELING QUERY DIFFERENCES IN QPP

The performance of dense coherence-based predictors is particularly accurate in certain dense retrieval settings (for TCT-ColBERT: pairRatio and A-pairRatio, for ANCE: WAND-embs and WD-embs) and shows superior performance for especially NDCG@10. Still, score-based predictors are often better for MAP@100. This difference in QPP correlations among evaluation metrics motivates us to explore whether the relationship between QPPs and retrieval effectiveness is mediated by the type of query (for instance queries of an Experience type have been found difficult to answer [6]). For this purpose, we apply a distribution-based QPP evaluation approach based on the scaled Absolute Rank Error (sARE) [20]. Specifically, the sARE value each query is calculated as: $sARE_{q_i} = \frac{|r_i^p - r_i^e|}{|Q|}$ where $r_i^p$ and $r_i^e$ are the ranks assigned to query $i$ by the QPP predictor and the evaluation metric, respectively (one sARE value is obtained per query, instead of a point estimate). This further allows using sARE in statistical models [18, 20]. Unlike [18, 20] who use ANOVA, we use *Linear Mixed Effects (LME)* models [13, 21, 32, 50], which also belong to Generalised Linear Models (GLM) [31, 35], but split the total explained variance in *sARE* into 2 levels.

Specifically, Level 1 specifies the within-query variations (how each query changes or the per query variance over different QPP predictors). Level 2 specifies the between-query differences; it further explains each part of Level 1 by showing, how it changes according to a between-query factor - here we use the type of query or *query type* as proposed in [6]. A 2-Level approach is necessary to model the interplay of QPPs with query types; while each query receives a separate sARE value for each QPP predictor, multiple queries in the same type share the same sARE, and are, therefore, nested within their group (each query belongs to only one level of query type). Thus, the multilevel approach allows splitting the total variation in sARE into within (due to QPPs - Level 1)- and between-query (due to query types - Level 2) variation. Using separate models for each evaluation measure allows to check which measure is more affected by query types. Next, we describe LMEs in detail.

**Table 3: Explanation of terms included in the linear mixed effects full model.**

| Parameter | Interpretation |
|---|---|
| Fixed effects | |
| $\gamma_{00}$ | average true sARE for the reference QPP predictor for the reference (without the effect of) query type |
| $\gamma_{01}$ | average difference in sARE between different query types for the reference QPP predictor |
| $\gamma_{10}$ | average true rate of change in sARE per unit change in QPP predictor for the reference (without the effect of) query type |
| $\gamma_{11}$ | average difference in sARE between different query types per unit change in QPP predictor |
| Random effects | |
| $\zeta_{0i}, \zeta_{1i}$ | allow individual true query trajectories to be scattered around the average query true change trajectory |
| $\epsilon_{ij}$ | allows individual query data to be scattered around individual query true change trajectory |
| Variance Components | |
| $\sigma_\epsilon^2$ | level 1 (residual) variance, variability around each query's true change trajectory |
| $\sigma_0^2, \sigma_1^1$ | level 2 variance in reference predictor and rate of change per predictor measurement, how much between-query variability is left after accounting for query type |
| $\sigma_{01}$ | residual covariance between true sARE for the reference (initial) predictor and rate of change, controlling for query type, across all queries |

## 6.1 Linear Mixed Model Definitions

First, our full model, denoted as $LME_{full}$, is defined as :
**Level 1**

$$sARE_{ij} = \pi_{0i} + \pi_{1i}(QPPPredictor) + \epsilon_{ij} \tag{8}$$

with $\epsilon_{ij} \sim N(0, \sigma_\epsilon^2)$
where $sARE_{ij}$ is the sARE of query $i$ at QPP predictor measurement $j$, $\pi_{0i}$ is the intercept (initial status) of query $i$'s change trajectory (reference QPP predictor, i.e., the first QPP measurement), $\pi_{1i}$ is the slope (rate of change) in sARE (per predictor unit), and $\epsilon_{ij}$ are the deviations of a query's equation on each measurement. This is also a way for Level 1 to check for statistically significant differences between predictors.

**Table 4: LMEs comparison and corresponding variance reduction type. Each row shows the $Pseudo - R^2$ of interest together with its definition.**

| Models compared | Quantity | Definition |
|---|---|---|
| $LME_{average}, LME_{QPP}$ | $Pseudo - R_\epsilon^2$ | $\frac{\sigma_{\epsilon_{LME_{average}}}^2 - \sigma_{\epsilon_{LME_{QPP}}}^2}{\sigma_{\epsilon_{LME_{average}}}^2}$ |
| $LME_{QPP}, LME_{Full}$ | $Pseudo - R_0^2$ | $\frac{\sigma_{0_{LME_{QPP}}}^2 - \sigma_{0_{LME_{full}}}^3}{\sigma_{0_{LME_{QPP}}}^2}$ |
| $LME_{QPP}, LME_{Full}$ | $Pseudo - R_1^2$ | $\frac{\sigma_{1_{LME_{QPP}}}^2 - \sigma_{1_{LME_{full}}}^3}{\sigma_{1_{LME_{QPP}}}^2}$ |

**Level 2**

$$\begin{cases} \pi_{0i} = \gamma_{00} + \gamma_{01}(QueryType) + \zeta_{0i} \\ \pi_{1i} = \gamma_{10} + \gamma_{11}(QueryType) + \zeta_{1i} \end{cases} \quad (9)$$

with $\begin{smallmatrix} \zeta_{0i} \\ \zeta_{1i} \end{smallmatrix} \sim MVN\left[\begin{bmatrix} 0 \\ 0 \end{bmatrix}, \begin{bmatrix} \sigma_0^2 \sigma_{01} \\ \sigma_{01} \sigma_1^1 \end{bmatrix}\right]$

where $\gamma_{00}$ and $\gamma_{10}$ are the average true sARE for the reference query type in the initial status and rate of change, respectively. Similarly, $\gamma_{01}$ and $\gamma_{11}$ show the effect of the between-query factor on sARE, for the initial status and rate of change. For convenience, we use $LME_{full}$ in an equivalent compact form (Levels 1 and 2) as:

$$sARE_{ij} = [\gamma_{00} + \gamma_{10}(QPPPredictor_{ij}) + \gamma_{01}(QueryType_i)$$
$$+\gamma_{11}(QueryType_i)(QPPPredictor_{ij})] \quad (10)$$
$$+[\zeta_{0i} + \zeta_{1i}(QPPPredictor_{ij}) + \epsilon_{ij}]$$

Table 3 shows the interpretation of each of the $LME_{full}$ parameters. Next, we introduce two reduced models. We start with $LME_{average}$ that only assumes an average sARE value:

$$sARE_{ij} = \gamma_{00} + \zeta_{0i} + \epsilon_{ij} \quad (11)$$

Finally, we obtain $LME_{QPP}$ as follows:

$$sARE_{ij} = \gamma_{00} + \gamma_{10}(QPPPredictor_{ij}) + \zeta_{0i} + \zeta_{1i}(QPPPredictor_{ij}) + \epsilon_{ij} \quad (12)$$

In what follows, we use a model selection strategy, as indicated in Table 4, where each row shows the models being compared, the quantity of interest, and its definition. The difference between $LME_{average}$ and $LME_{QPP}$ is the effect of QPP predictor; $Pseudo - R_\epsilon^2$ tells us how much of the total variability within queries can be attributed to QPPs. Similarly, when comparing $\sigma_0^2$ and $\sigma_1^2$ of $LME_{full}$ with the ones of $LME_{QPP}$, these two models differ in the inclusion of the terms $\gamma_{01}(QueryType)$ and $\gamma_{11}(QueryType)$. $Pseudo - R_0^2$ and $Pseudo - R_1^2$ tell us how much of the total variability between queries in initial status and rate of change, respectively, are due to query type. Starting from $LME_{average}$, we sequentially move to $LME_{QPP}$ and $LME_{full}$, if needed. At each step, we compare between the model that contains the added factor and the one that does not. The decision is made based on the significance of fixed effects and the model Deviance [32, 50], indicating the goodness-of-fit (the lower, the better). The deviance in this case is: $Deviance = -2LL_{Max}$, where $LL_{Max}$ is the maximised log-likelihood of each model. We implement the proposed LMEs using the lme4 R package [4, 52], with Full Maximum Likelihood Estimation. We now address the following research questions:

**RQ3** Is the accuracy of query performance prediction influenced by query type more for dense retrieval than sparse retrieval?

**RQ4** How sensitive are the different evaluation measures to (a) query types and (b) QPPs?

## 6.2 RQ3 - Importance of Query Type

Table 5 provides the resulting LMEs from our model comparison strategy, as outlined in Section 6.1. For the dense retrieval models, Equations with $sARE_{MAP}$ contain a coefficient that indicates sensitivity to a particular type of query, (the first line of ANCE refers to Not-A-Question queries, and the first two lines in TCT-ColBERT refer to Experience and Reason queries). The corresponding BM25 LMEs do not contain a query type coefficient).

Most importantly, in Table 6, the top half of shows the proportions of gained explained variance for both levels (with ✗ indicating no significant gains), while the bottom half highlights the included effect terms. The first row shows that variations due to QPPs are similar for the three retrieval methods (similar $Pseudo - R_\epsilon^2$ values). However, the next two rows have much higher relative gain in explained variance for the two dense models than BM25, especially for $Pseudo - R_1^2$, reaching 35% and 23% for ANCE and TCT-ColBERT, respectively. Indeed, as $Pseudo - R_1^2$ includes query type, this means that a noticeable proportion of the variance is attributed to query type. Therefore, for dense retrieval, some query types are more accurately predicted by certain QPPs, and other query types work better for other QPPs. This indicates that QPP performance cannot be judged in isolation from query taxonomies, which in some cases are more influential than the predictor itself. To answer RQ3, the accuracy of query performance is influenced by query type more for dense retrieval than sparse retrieval.

## 6.3 RQ4 - Sensitivity of Evaluation Measures

Figure 3 plots the TCT-ColBERT $LME_{Full}$ of sARE prediction for both $sARE_{MAP}$ (a) and $sARE_{NDCG}$ (b). In each plot, the sARE (y-axis) values are plotted as a function of QPP predictor (x-axis), with each query type as a separate plot, and colouring indicate different QPP predictors (from left: starting with dense coherence-based predictors, then supervised, and score-based on the right). For $sARE_{MAP}$, the trends for two query types, Experience and Reason, behave differently than the rest; these two types show better performance (lower sARE) for coherence-based than score-based predictors, while the opposite holds for Instruction and Not-A-Question queries. As for Evidence-based and Factoid queries, there is higher variance in sARE among different queries, but for dense coherence-based predictors, the variance is smaller than score-based predictors, as indicated by the corresponding colours. In general, for $sARE_{MAP}$, performance seems to be affected by the different types of queries, which make QPPs more unstable. Indeed, Experience and Reason were found as *harder* questions in the original categorisation study [6]. This result reflects the selected model for $sARE_{MAP}$, which was $LME_{Full}$ (effect of query type across QPP measurements).

On the other hand, for $sARE_{NDCG}$, QPP performance for different query types seems more uniform. The trend still looks different for Experience and Not-A-Question queries compared to the rest, but those represent only a small portion of the total queries. For the remaining types, the structure is similar, with some variations in strength. Importantly, for Evidence-based, Factoid, Instruction, and Reason queries, there is increasing variance across queries for

**Table 5: Resulting LME models for each retrieval method and all metrics.**

| BM25 | |
|---|---|
| $sARE_{MAP}$ | $sARE_{ij} = [0.29 - 0.009(QPPPredictor_{ij})] + [\zeta_{0i} + \zeta_{1i}(QPPPredictor_{ij}) + \epsilon_{ij}]$ |
| $sARE_{NDCG}$ | $sARE_{ij} = 0.26 + \zeta_{0i} + \epsilon_{ij}$ |
| $sARE_{MRR}$ | $sARE_{ij} = 0.30 + \zeta_{0i} + \epsilon_{ij}$ |
| **ANCE** | |
| $sARE_{MAP}$ | $sARE_{ij} = [0.28 - 0.008(QPPPredictor_{ij}) + 0.25(NotAQ_i) + 0.05(NotAQ_i)(QPPPredictor_{ij})] + [\zeta_{0i} + \zeta_{1i}(QPPPredictor_{ij}) + \epsilon_{ij}]$ |
| $sARE_{NDCG}$ | $sARE_{ij} = 0.25 + \zeta_{0i} + \epsilon_{ij}$ |
| $sARE_{MRR}$ | $sARE_{ij} = [0.35 - 0.008(QPPPredictor_{ij})] + [\zeta_{0i} + \zeta_{1i}(QPPPredictor_{ij}) + \epsilon_{ij}]$ |
| **TCT-ColBERT** | |
| $sARE_{MAP}$ | $sARE_{ij} = [0.32 - 0.01(QPPPredictor_{ij}) + 0.05(Experience_i)(QPPPredictor_{ij})] + [\zeta_{0i} + \zeta_{1i}(QPPPredictor_{ij}) + \epsilon_{ij}]$ |
| $sARE_{MAP}$ | $sARE_{ij} = [0.32 - 0.01(QPPPredictor_{ij}) + 0.02(Reason_i)(QPPPredictor_{ij})] + [\zeta_{0i} + \zeta_{1i}(QPPPredictor_{ij}) + \epsilon_{ij}]$ |
| $sARE_{NDCG}$ | $sARE_{ij} = [0.32 - 0.008(QPPPredictor_{ij})] + [\zeta_{0i} + \zeta_{1i}(QPPPredictor_{ij}) + \epsilon_{ij}]$ |
| $sARE_{MRR}$ | $sARE_{ij} = 0.32 + \zeta_{0i} + \epsilon_{ij}$ |

**Table 6: Proportion of explained variance per component and included fixed effects in each LME for all three retrieval methods. ✓ indicates the presence of a fixed effect in LMEs, while ✗ shows the absence of either an important contribution of a factor (top) or a fixed effect (bottom).**

| | BM25 | | ANCE | | TCT-ColBERT | |
|---|---|---|---|---|---|---|
| sARE → | MAP | NDCG | MAP | NDCG | MAP | NDCG |
| $Pseudo - R_\epsilon^2$ | 13.4% | ✗ | 7.5% | ✗ | 12.4% | 14.6% |
| $Pseudo - R_0^2$ | ✗ | ✗ | 17.2% | ✗ | 2.2% | 9.9% |
| $Pseudo - R_1^2$ | ✗ | ✗ | 35.6% | ✗ | 22.8% | 8.1% |
| $\gamma_{00}$ | ✓ | ✓ | ✓ | ✓ | ✓ | ✓ |
| $\gamma_{01}$ | ✗ | ✗ | ✓ | ✗ | ✓ | ✗ |
| $\gamma_{10}$ | ✓ | ✗ | ✓ | ✗ | ✗ | ✓ |
| $\gamma_{11}$ | ✗ | ✗ | ✓ | ✗ | ✓ | ✗ |

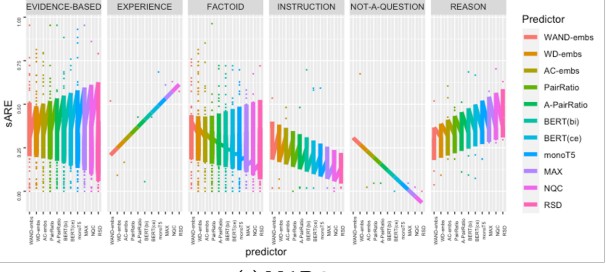

**(a) MAP@100**

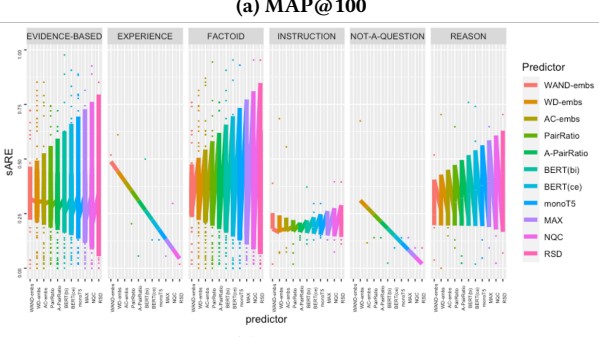

**(b) NDCG@10**

**Figure 3: LME results from the full model for TCT-ColBERT.**

score-based compared to dense coherence-based predictors. This indicates that our proposed predictors are less sensitive to query

type compared to score-based and supervised predictors. Note that while we plot the full model, for $sARE_{NDCG}$, $LME_{QPP}$ was preferred, i.e., only an effect of QPP predictor. This is complemented by Table 5, where $sARE_{NDCG}$ contain a coefficient for QPPs, but not for query types or their interaction with QPPs.

To summarise, in Section 5.1, we observed that score-based predictors showed improved performance for MAP@100, but our LME analysis showed that this result is susceptible to influential query types. Instead, our dense coherence-based predictors showed higher correlations mainly for NDCG@10, and with the LME analysis (lack of query type terms and $Pseudo - R^2$ terms at Level 2), we showed that this is more stable across different query types. Therefore, our predictors provide promising evidence for generalisability compared to existing predictors. In other words, while both MAP@100 and NDCG@10 are sensitive to QPPs, NDCG@10 is less sensitive to query type variations than MAP@100, thereby answering RQ4.

## 7 CONCLUSIONS

We examined the accuracy of QPP upon two single-representation dense retrieval methods (ANCE and TCT-ColBERT). In particular, we proposed new variants of unsupervised coherence-based predictors and managed to increase their performance for dense retrieval. In this way, we showed that changing the representations from TF.IDF to neural embeddings provided by the dense retrieval models together with some further modifications is enough to generalise performance of unsupervised predictors in relation to supervised ones. Indeed, with increasing effectiveness brought by dense retrieval methods, our proposed predictors becomes more competitive, especially for NDCG@10 and MRR@10. Also, we highlighted that focusing on a single evaluation measure to optimise a proposed predictor can be problematic and may falsely inform future studies, since MAP@100 and NDCG@10 cannot be used interchangeably. At the same time, we demonstrated the interplay between the different QPP predictors, evaluation metrics, and the particular types of queries. Importantly, we showed that while score-based predictors still remain very competitive for MAP@100, our examined statistical models indicate that MAP@100 is highly influenced by the type of query. Instead, using NDCG@10, QPP performance is more stable across queries, and since our proposed predictors show higher performance on this metric, this is a promising result for more generalisable performance in dense retrieval.

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
