# OpenReview forum: "Coherence-based Query Performance Measures for Dense Retrieval"
_ACM.org/SIGIR/ICTIR/2024/Conference — ICTIR 2024_

### Official Review · Reviewer_JH1Q · 2024-05-09

**Rating:** 0
**Confidence:** 4

**Objective Part Of Review:**

The paper conduct a comprehensive study on query performance measures for dense retrieval. The motivation is clear. The authors indeed provide an improved coherence-based approach. The main concern for me is that the authors claim their coherence-based are better than score-based approaches (especially in NDCG@10); however, my observation is that score-based approaches are still better overall even when just looking at NDCG@10. I’m not an expert in the field of QPP but I feel maybe combine the score-based and the proposed coherence-based approaches can even reach better results and make the paper’s claim more convincing.

**Subjective Part Of Review:**

I think overall, the arrangement of the paper is good enough and make the paper easy to follow. The main point unclear to me is section 3.1.1. I cannot understand the equation (2). There is no explanation for $\vec{d}$ and the operator *.  In addition, I find it not convinced to me with the authors’ conclusion that query performance is influenced by query type more for dense retrieval than sparse retrieval. The paper only compares between neural dense retrieval and traditional sparse retrieval. It may make sense that neural dense retrieval shows higher variance since it high ranking performance. I’m not sure if we can get the same conclusion if we compare to neural sparse retrieval, such as uniCOIL and SPLADE.

---

### Official Review · Reviewer_KqV8 · 2024-05-14

**Rating:** -1
**Confidence:** 5

**Objective Part Of Review:**

The paper addresses the query performance prediction task for dense retrieval. A few measures of coherence of the retrieved list, measured via similarities of embedding vectors, are proposed.

The paper is well written in general (except for some minor issues) and relatively easy to follow. The motivation is good (see more details below) and the coverage of related work is also good.

**Subjective Part Of Review:**

The suggested measures are heuristic in nature. There is no underlying theory but rather (good intuition) which draws much inspiration from previous work as coherence based predictors were extensively studied for sparse retrieval.

The performance numbers are mixed with no clear signal about the superiority of the suggested measures to existing ones (e.g., NQC and RSD). ICTIR is not about the performance numbers but rather about the underlying formalism and theory as I view it. However, given that the suggested measures are not significant extension of previously proposed measures, nor are they based on some theoretical framework, one would expect to see clear value with the performance numbers so as to draw conclusions about what works and what doesn't for sparse retrieval. Indeed, studying qpp for sparse retrieval is an important question and I agree with the authors that BERT-qpp methods do not seem to be the way to go here.

I also have an issue with the fact that Pearson or Spearman were not used because Kendall's tau doesn't account to ranks and top ranks are more important. I read the authors' note about why they haven't used Pearson and Spearman but wasn't convinced that they shouldn't be presented, especially, given that this was the state-of-affairs in many previous reports.

As a final micro-level comment, it is not clear with respect to what statistical significance tests have been performed.

---

### Official Review · Reviewer_Y8gy · 2024-05-17

**Rating:** 0
**Confidence:** 4

**Objective Part Of Review:**

This paper study (score/coherence/supervised-based) and propose (coherence-dense) QPP metrics for evaluating dense models (ANCE and ColBERT) on three metrics (MAP, nDCG, and MRR). The authors propose to study the effect of the query type (but do not say how this query type is determined), and show (properly) that it is a very important factor given the lack of precision of current QPP indicators (thus showing, although this is not stated by the authors, that there is a lot of progress that could be made on this side).

In section 3.2.3, the authors propose "pairRatio" which is interesting since it underlies the fact that ranking is not really taken into account by coherence-based models. Note that the problem of computing a rank-biased correlation metric has been tackled in "A new rank correlation coefficient for information retrieval" (SIGIR 2008) with a more principled (but quite different) approach.

There are many unclear parts and typos in the paper:

- introduction: one explanation (of what?)
- figure 1: the figure intent is not very clear
- the research questions could be stated in the introduction (rather than spread into the paper)
- "one explanation could be": explanation of what?
- eq.2: I believe $*$ is for matrix product (which is not standard)
- $W_Phi$ is noted $W^\phi$
- what is the "the (diagonal) subset of $W^\phi$?
- table 3: the formatting does not help in matching the two columns
- (section 6) |Q| is not defined
- section 6.1: what is the QPP predictor measurement? the predictor unit? the change trajectory? MVN ?
- is table 5 needed?
- figure 3 (and the text): it seems that the reason for the query type impacting differently differently the performance (e.g. experience / not a question) this is mainly due to the small number of samples. This should be clarified in the text.
- section 6.2: how are query types determined?

**Subjective Part Of Review:**

More generally, this paper does not deliver a very clear message, and it seems the author change their mind during writing: the paper oscillates between proposing new solutions (e.g. the vector-based adaptations of coherence-based QPP metrics) and the analysis (e.g. unsupervised vs supervised) – especially since the improvement of the proposed method is only on one metric out of the three (the score-based QPP is performing better in most of the cases). I think there is an effort to be made so that the message the authors want to deliver is clearer.

Finally, most of the value of the paper lies in the evaluation of many QPP metrics for different models (but here, more evaluated neural models would have been needed for more important conclusions).

---

### Official Review · Reviewer_ZN4X · 2024-05-17

**Rating:** 0
**Confidence:** 3

**Objective Part Of Review:**

The authors propose unsupervised coherence-based predictors for dense retrieval. Overall, the paper is well-structured, and the main results are presented properly. My major concern is whether the main claim that "the proposed method managed to increase their performance for dense retrieval" holds.

I believe methods such as AC-embs should be considered more as baselines because they compute document similarities based on the document representation used in the retrieval system, similar to their use in sparse retrieval. The novel aspect is the proposed pairRatio; however, I suspect that different values of k will significantly affect the prediction, as the appropriate bottom left corner for each query is uncertain. From the main results table, pairRatio does not outperform score-based methods by itself in most cases unless in the adjusted method that also considers query-doc scores.

**Subjective Part Of Review:**

It is appreciated that the authors conducted a statistical analysis to understand the importance of query type. However, more clarification on how this was done is needed. For example, is it still based on DL19 and DL20? If so, how were the query types classified, and how many samples were used for the analysis? I am not convinced that ~50 queries from TREC DL are sufficient for robust modeling.

A more detailed introduction on the motivation of post-retrieval QPP and how it benefits a retrieval system would be beneficial.
Overall, I think this work is within the interest of ICTIR, and it is interesting to see the corresponding statistical analysis, although some concerns remain as stated above.


typo sec 5.1: MRRR -> MRR

---

### Meta-Review · Area_Chair_eZAS · 2024-05-20

**Recommendation:** Reject
**Confidence:** 4

**Metareview:**

The paper addresses an important question in QPP for sparse retrieval and includes interesting statistical analyses. Nevertheless, this paper requires more clarity, conducts more baseline experiments, and provides a stronger theoretical demonstration.
1.  This paper should be revised. There are so many typos and format errors. As mentioned by reviewers, lots of the details are unclear, such as images and formulas. More experimental details should also be added.
2. The paper's primary contribution may lie in its evaluation of various QPP metrics across different models. However, the inclusion of more neural models in the evaluation could have led to more substantial conclusions. Additionally, the suggested measures seem heuristic without a solid theoretical foundation.
3. I suggest that this paper should be revised and resubmitted.